# Pooling region learning of visual word for image classification using bag-of-visual-words model

Ye Xu[1]*, Xiaodong Yu[2], Tian Wang[2], Zezhong Xu[2]

**1** School of IoT Technology, Wuxi Institute of Technology, Wuxi, China, **2** School of Computer Information and Engineering, Changzhou Institute of Technology, Changzhou, China

* xuye@wxit.edu.cn

## Abstract

In the problem where there is not enough data to use Deep Learning, Bag-of-Visual-Words (BoVW) is still a good alternative for image classification. In BoVW model, many pooling methods are proposed to incorporate the spatial information of local feature into the image representation vector, but none of the methods devote to making each visual word have its own pooling regions. The practice of designing the same pooling regions for all the words restrains the discriminability of image representation, since the spatial distributions of the local features indexed by different visual words are not same. In this paper, we propose to make each visual word have its own pooling regions, and raise a simple yet effective method for learning pooling region. Concretely, a kind of small window named observation window is used to obtain its responses to each word over the whole image region. The pooling regions of each word are organized by a kind of tree structure, in which each node indicates a pooling region. For each word, its pooling regions are learned by constructing a tree with its labelled coordinate data. The labelled coordinate data consist of the coordinates of responses and image class labels. The effectiveness of our method is validated by observing if there is an obvious classification accuracy improvement after applying our method. Our experimental results on four small datasets (i.e., Scene-15, Caltech-101, Caltech-256 and Corel-10) show that, the classification accuracy is improved by about 1% to 2.5%. We experimentally demonstrate that the practice of making each word have its own pooling regions is beneficial to image classification task, which is the significance of our work.

## 1 Introduction

Image classification, as one of the most challenging tasks in computer vision, has attracted much attention in decades. Its target is to classify images into semantic predefined classes. There are many challenges in image classification task, such as the change in viewpoint, illumination, partial occlusion, clutter, inter and intra-class visual diversity. A great number of works have proposed to deal with these challenges. Nowadays, two popular classification

**Data Availability Statement:** All data sets are available from Figshare. (Scene-15: https://figshare.com/articles/Scene-15/12103434; Caltech-101: https://figshare.com/articles/Caltech-101/12103614; Caltech-256: https://figshare.com/

**Funding:** The author(s) received no specific funding for this work.

**Competing interests:** The authors have declared that no competing interests exist.

frameworks, Bag-of-Visual-Words (BoVW) and convolutional neural network (CNN), have achieved many encouraging results in the past ten years.

BoVW model is a conventional yet effective technique applied in the fields of image classification, image retrieval and object recognition. In BoVW model, local features such as scale-invariant feature transform (SIFT) [1] are extracted from a set of images to learn a visual dictionary consisting of visual words. Subsequently, local features are encoded as the coding vectors using the learned dictionary. For each local feature, the non-zero coefficients of its coding vector indicate which visual words involve in the coding process. At last, all the coding vectors are aggregated into one vector, namely the image representation vector, by performing a pooling operation such as max pooling or average pooling [2]. CNN [3] is a deep neural network of exploiting the spatial structure of image. It consists of convolutional layers, pooling layers, non-linear activations, and fully connected layers. Convolutional layer reflects the existences of various shapes, which are detected by different convolutional kernels. Pooling layer retains the maximum saliency in local region by down sampling convolutional layer. Fully connected layers are generally appended at the end, which simply represent the multi-layer perceptron. Non-linear activation functions are necessary to learn complex functions. A predominant difference between BoVW and CNN is that, BoVW works with hand-crafted features such as SIFT and Histogram of Oriented Gradient (HOG), while CNN can extract automatically image features after training it on a significant amount of data.

Compared with BoVW, CNN shows significant performance on many challenging image datasets, such as ImageNet Large-Scale Visual Recognition Challenge (ILSVRC) [4]. However, when dealing with small dataset with limited training images, directly learning CNN on the limited data shows a poor performance due to over-fitting [5]. A significant amount of data to train is indispensable for learning an applicable CNN. To solve this, transfer learning [6] can be used to make the training of CNN on small dataset feasible by employing the weights from a pre-learned CNN. The first convolutional layers of a pre-learned CNN are adopted as mid-level feature extractor, and its rest layers are modified to fit target task. When training the adapted CNN, only the weights of the modified layers are updated, while the weights of other layers are fixed. Some methods [7] [8] based on transfer learning have reported obvious improvement over BoVW, and achieved significant results in medical image classification [9].

Despite the significant effectiveness of transfer learning, it needs a pre-learned CNN, which requires huge amounts of data (in millions) and time for training. Besides, the number of the parameters of a pre-learned CNN is also huge, resulting in considerable memory space consumption, such as about 520M for VGG-16 [10]. Another problem is that, the choice of the source dataset used for pre-learning a CNN and the number of the images in target dataset influence the classification result [11]. Negative transfer is likely to happen if the source dataset is completely unrelated to target dataset. Thus, it is not very clear on whether target task benefits from a pre-learned CNN. At the same time, BoVW model is a plug-n-play method which can be adopted without any prior initialization or training [12]. In the problems where there is not enough data to train CNN, or it is hard to apply a pre-learned CNN due to the memory limitation or find an applicable pre-learned CNN, BoVW is still a good alternative for image classification. In consequence, we advocate the conventional yet effective BoVW in this paper.

In BoVW model, the spatial distribution of local feature has close relationship with image class. Many works have devoted to incorporating the spatial information of local feature into the final image representation. Nevertheless, almost all the works do not make each word have its own pooling regions by considering the difference among the spatial distributions of the local features indexed by different words. As illustrated by Feng et.al [13], for images from a specific class, their local features indexed by the same word often share similar spatial distribution, and class-specific spatial distributions are distinct from each other. In [13], the authors

learned a weighted $l_p$-norm spatial pooling function tailored for the class-specific feature spatial distribution. However, this pooling function is still used under the framework of spatial pyramid matching (SPM) [14].

In this paper, we propose to make each word have its own pooling regions, and raise a simple yet effective method for learning pooling region. Specifically, a kind of small window named observation window is used to obtain its responses to each word over the whole image region. We adopt a kind of tree structure to organize the pooling regions of each word. The pooling regions of each word are learned by constructing a tree with the labelled coordinate data. The labelled coordinate data consists of the coordinates of responses and image class labels. In the process of tree construction, when dividing a pooling region of parent node, we employ linear discriminant analysis (LDA) to learn a dividing direction, and select the best dividing line from the set of the candidate dividing lines in this direction according to information gain. The effectiveness of our method is validated by observing if there is an obvious classification accuracy improvement after applying our method. Our experiments are conducted on four small datasets, i.e., Scene-15 [14], Caltech-101 [15], Caltech-256 [16] and Corel-10 [17]. Our experimental results show that the classification accuracy is improved by about 1% to 2.5%. This phenomenon demonstrates that the practice of making each word have its own pooling regions, is beneficial to image classification task.

The remainder of this paper is organized as follows: the proceeding section is about the related works. Section 3 illustrates our work in detail. Experimental evaluation and analysis are reported in Section 4, and the conclusion is drawn in Section 5.

## 2 Related works

The most related work is SPM proposed by Lazebnik et al. [14]. It partitions the whole image into multiple blocks at different resolution levels of $1 \times 1$, $2 \times 2$ and $4 \times 4$, and then concatenates the pooling vectors obtained in these blocks to form the image representation vector. Some improved works are proposed based on SPM. Huang et al. [18] weighted the spatial locations of local features from each block by a Gaussian function. Wu et al. [19] built a directed graph by viewing the blocks as the nodes to consider the relationship between the blocks. Harada et al. [20] proposed to form the image representation vector as a weighted sum of all the pooling vectors. Considering that SPM is not invariant to global geometric transformation, some works have devoted to solving this problem. Zhang et.al [21] proposed different heuristic methods by employing three frequency histograms, i.e. shapes, pairs and binned log-polar features representation. Penatti et al. [22] proposed an method named word spatial arrangement (WSA). It captures the relative positions of visual words by partitioning the image space into four quadrants making the position of a given word as the origin, and then aggregates the statistics of all the words from each quadrant. Except for these works devoting to dividing images into subregions of different shapes, some works focus on learning discriminative image regions, such as [23] and [24]. In [23], a saliency map indicating the discriminability of local feature is used to weight the coding vectors of local features. In [24], a latent support vector machine is adopted to learn a set of latent pyramidal regions. Jia et al. [25] proposed to adaptively learn the discriminative blocks from a set of overcomplete spatial blocks, and a boosting method to block learning is introduced by Zhang et al. [26].

Another way of incorporating the spatial information is to encode relationship or cooccurrence of visual words. The works [27] [28] [29] group the spatially close visual words into visual phrases and then represents an image as a histogram of these phrases. Similarly, Silva et. al [30] and Dammak et.al [31] employed graph instead of visual phrase to accurately describe the spatial relationships among visual words. Different from obtaining visual phrases after

feature quantization, Morioka et al. [32] and Boureau et al. [2] concatenated the neighboring local features into a joint feature to preserve the local region information. A notable work proposed by Khan et al. [33] considered the global geometric relationships among the Pairs of Identical Words (PIWs). Based on the angles between these identical visual words, a normalized histogram is calculated termed as PIWAH (Pairs of Identical Visual Word Angle Histogram). Anwar et al. [34] extended this work to encode the global geometric relationships of the visual words in a scale- and rotation-invariant manner. This work computes angles made by triplets of identical visual words, and then constructs histograms from these angles termed as TIWAH (Triplets of Identical Visual Words Angle Histogram). Based on this work, a very recent work proposed by Zafar et al. [35] calculates an orthogonal vector relative to each point in the triplets of identical visual words and then created the histogram from the magnitude of these orthogonal vectors.

## 3 Our work

In this section, we first illustrate our work under the framework of BoVW model. Afterwards, the detail about observation window is presented. Next, the method for learning pooling regions is illustrated in detail. In the end, we explain how to obtain the image representation vector.

### 3.1 Process of image representation

BoVW model has formed a unified framework over the last decade. There are five basic stages in this framework. These stages are image patch extraction, image patch description, dictionary learning, feature coding and feature pooling, respectively. Our work only involves in the last stage, i.e., feature pooling. Fig 1 shows the process of image representation including our work.

As shown in Fig 1, the input image is converted into the set of the coding vectors through the first four stages. The first stage is to extract patches from the input image. This process is implemented via sampling local areas of the image usually in a dense manner, e.g., the dense patches of $16 \times 16$ pixels with the step of 8 pixels. Then, the image patches are described as the feature descriptors (local features). This process is usually implemented via statistical analysis over pixels of image patches. SIFT is widely used to discribe image patch as a 128-dimensional vector. Except for SIFT, local binary pattern and HoG are also employed in some works. Afterwards, the feature descriptors are encoded as the coding vectors with a visual dictionary, which is generated using the feature descriptors extracted from all training images. Each feature descriptor activates a number of visual words, and generates a coding vector, whose length is equal to the number of visual words. The difference of various coding methods lies in how to activate the visual words.

At the last stage, the set of the coding vectors are converted into an image representation vector by our proposed method. Concretely, this stage consists of three steps: 1) obtaining the responses of different kinds of observation windows to each visual word (illustrated in Section 3.2); 2) grouping the responses to each visual word in terms of the pooling regions learned for the word (illustrated in Section 3.3); 3) taking the maximum from each group, and concentrating all the maximums to form the image representation vector (illustrated in Section 3.4).

### 3.2 Observation window

In this paper, we propose the observation window. For clarity, we illustrate the principle of observation window under the condition that each feature descriptor is represented only by

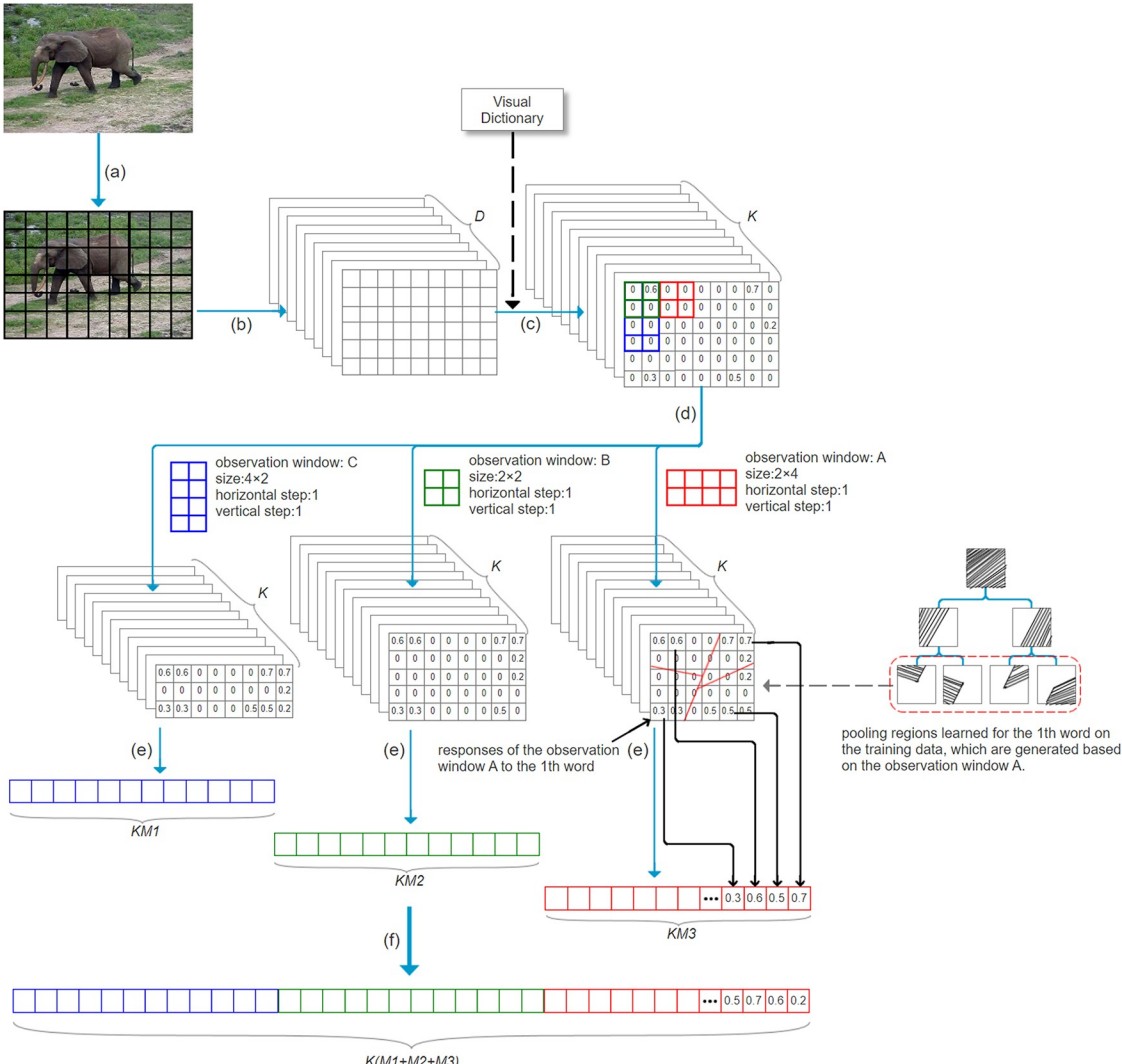

**Fig 1. Process of image representation.** (a) extracting the image patches; (b) describing the image patches as the $D$-dimensional feature descriptors; (c) encoding the descriptors as the $K$-dimensional coding vectors with a visual dictionary including $K$ visual words; (d) obtaining the responses of different kinds of observation windows to each word; (e) grouping the responses to each word in terms of the pooling regions learned for the word; (f) taking the maximum from each group; (g) concentrating all the maximums to form the image representation vector.

the most similar visual word to it. An observation window of the size $(w, h)$ includes $w \times h$ image patches. If the feature descriptor of some image patch in an observation window is represented by the $i$th visual word, the response of the window to the $i$th word is set 1 from 0 to denote that the $i$th word exists in the window. The observation window is placed on image by certain horizontal and vertical steps to obtain the responses of the window to each word over the whole image region. For each word, the responses of observation window to it form a response matrix. Fig 2 explains this process. Furthermore, the observation windows of different sizes and steps can also be applied simultaneously, as shown in Fig 1. In this case, for each kind of observation window, each word has a corresponding response matrix obtained with the window. Instead of the coding vectors, the response matrices of all the words are used to obtain the image representation vector.

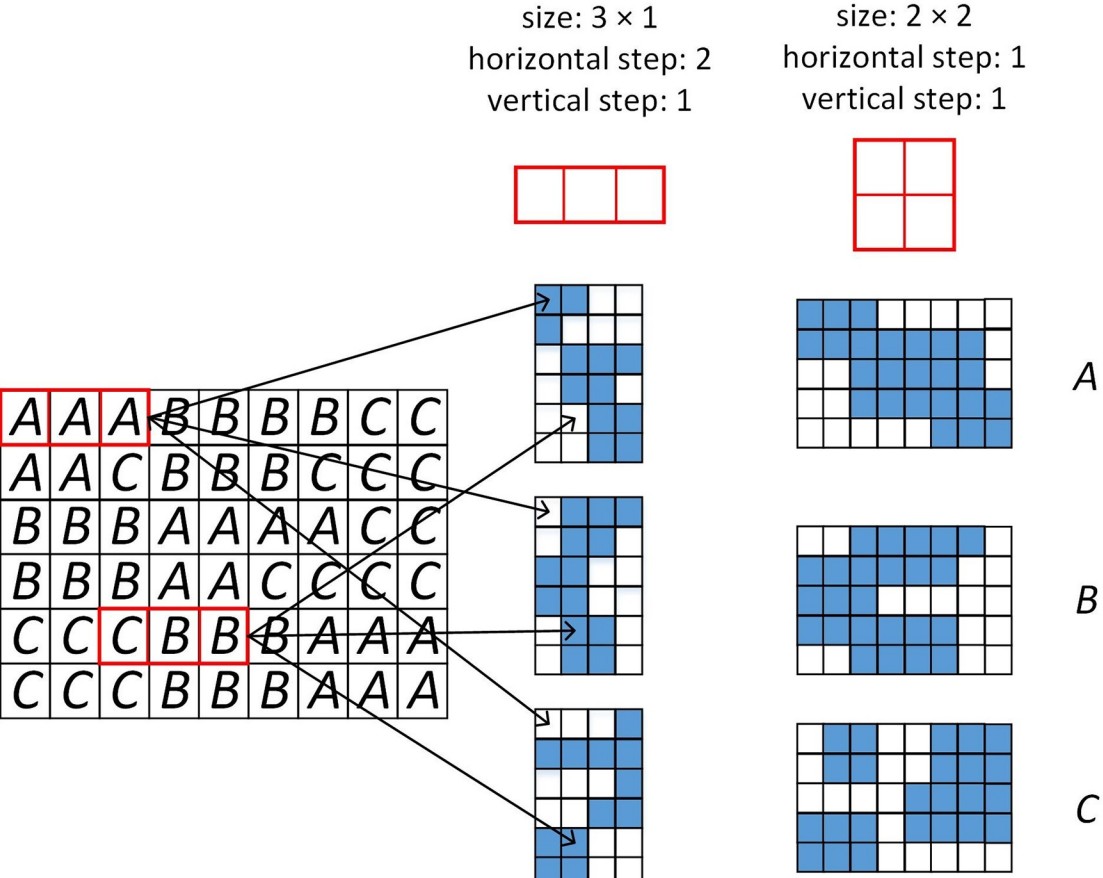

**Fig 2. Principle of observation window.** Three visual words A, B and C are used to encode the feature descriptors from the left image. The right part shows the existences (denoted by the blue blocks) of these words in the two kinds of observation windows.

In practice, feature descriptor is usually encoded by multiple visual words. It is assumed by our method that in the place where feature descriptor locates at, multiple visual words appear simultaneously. In this case, the response of an observation window to the $i$th visual word is the maximum of the coding coefficients of the feature descriptors (included in the window) to the $i$th word.

The advantages of using observation window are twofold. One is that, the robust of image representation to image variability, such as object size, location and poses, is improved by checking the existence of visual word in a slightly larger window (demonstrated in Tables 2 and 3). Another is that, the existences of visual word in the windows of different sizes and steps provide more discriminative information than in the window of single size. In fact, this observation is supported by the effectiveness of SPM. SPM partitions an image into multiple blocks at different resolution levels of $1 \times 1$, $2 \times 2$ and $4 \times 4$. From the viewpoint of feature descriptor, for each descriptor, the existence of the word assigned to it is checked in three kinds of windows ($1/4 \times 1/4$, $1/2 \times 1/2$ and $1 \times 1$ image size).

### 3.3 Pooling regions of visual word

A pooling region is a 2-dimensional region in image space. In the existing methods (e.g., [14, 18, 19]), the whole image region is divided into multiple pooling regions. The coding vectors

of all the feature descriptors from an image are grouped by the pooling regions. The coding vectors in a group are aggregated into a pooling vector by computing a statistical value (e.g., maximum value) of the coding coefficients to each visual word, respectively. The length of the pooling vector is equal to the number of visual words. From the viewpoint of viusal word, the coding coefficients to each word are grouped by the same pooling regions, and a statistical value is computed for each group.

In this paper, we allow each word have its own pooling regions. In this case, the coding coefficients to each word can be grouped by its own pooling regions. There is a reason for this practice. Feng et.al [13] have pointed out that, for images from a class, their feature descriptors indexed by different words have distinct spatial distributions, and their feature descriptors indexed by same word often share similar spatial distribution. Besides, class-specific spatial distributions are distinct from each other. This observation implies that, 1) each word can have its own pooling regions. These regions can be designed more discriminative; 2) the pooling regions of different shapes, sizes and locations have different discriminability.

By taking into account the above fact, we propose to learn its pooling regions for each visual word in terms of its spatial distributions on the images of different classes. To achieve this, the pooling regions of visual word are organized by a kind of tree structure, as shown in Fig 3, Each node corresponds to one pooling region indicated by the blue colour. The shape of region is not restrained to the rectangular shape by allowing that the region of parent node can be divided in any direction. In such a tree, the root node indicates the whole image region, and each parent node has two child nodes, which correspond to two subregions. The combination of the pooling regions indicated by all the nodes at any level is the whole image region. In this manner, SPM can also be easily represented by this kind of tree structure. The regions indicated by the nodes at the levels 0, 2, 4 correspond to the regions divided at the resolution levels of $1 \times 1$, $2 \times 2$ and $4 \times 4$.

Based on the tree-based pooling region representation, the pooling regions can be learned by constructing a tree from the root node to the leaf nodes. The key of pooling region learning is how to divide the pooling region of a parent node. Moreover, it is worthing note that, for each word, instead of the coding coefficients to it, the responses of observation window to it need to be grouped by its own pooling regions in our method. To this end, we generate the labelled coordinate data using the coordinates of the responses of observation window and image class labels (illustrated in Section 3.3.1), and learn the best dividing line using the labelled coordinate data (illustrated in Section 3.3.2). After obtaining a tree, we group the responses of observation window by the pooling regions indicated by the nodes of the tree (illustrated in Section 3.3.3).

**3.3.1 Labelled coordinate data.** In our method, the pooling regions of a visual word are applied on the responses of observation window to it to group the responses. Therefore, for each word, its pooling regions need to be learned in terms of its spatial distributions obtained with observation window on the training images of each class. The class-specific spatial distribution data of a visual word can be attained by recording the coordinates of the non-zero responses to it on each training image and the class label of this image. The coordinate of a response is defined as the average value of the center coordinates of the image patches included by the observation window that generates the response.

Concretely, a coordinate $s = (x, y)$ of response and a class label $c$ constitute a labelled coordinate datum $(s, c)$. For the $k$th word, the coordinates of the non-zero responses to it on the $j$th image and the class label $c_j$ of this image are combined to form the labelled coordinate data $B_j^k = \{(s_i^j, c_j), i = 1, 2, ..., N_j\}$ of the word obtained on the $j$th image, where $N_j$ is the number of the non-zero responses to the $k$th word. The labelled coordinate data $B_j^k$

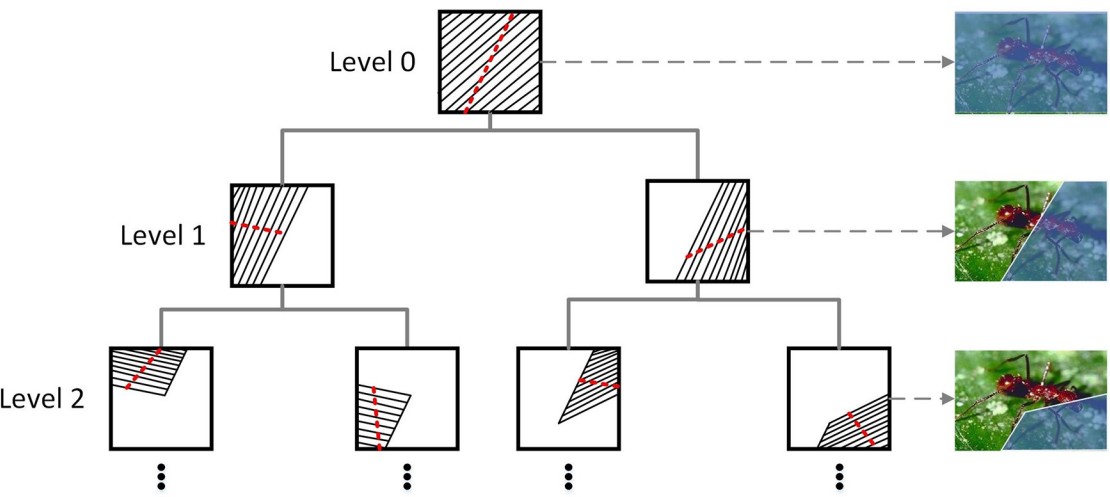

(a) Tree-based pooling region representation

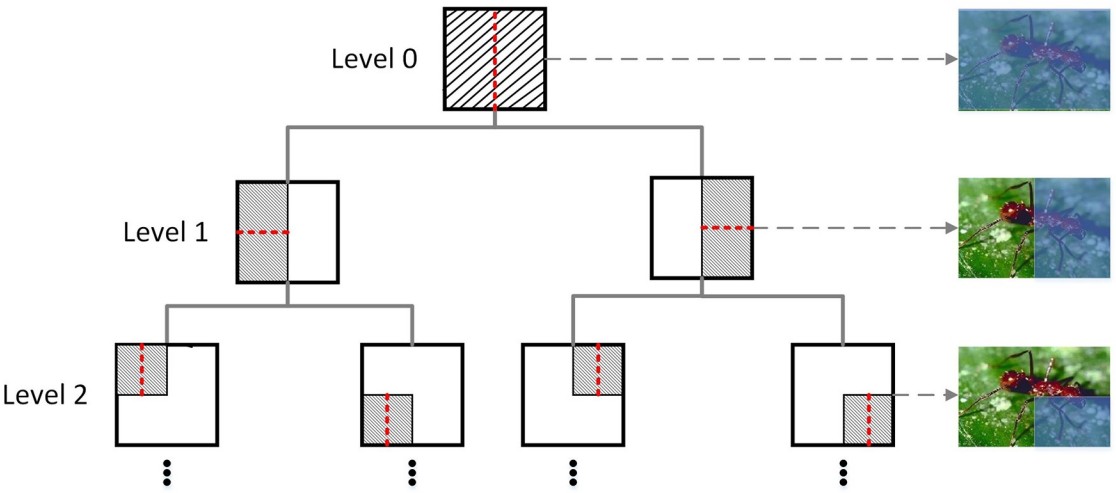

(b) Pooling regions of SPM represented by a kind of tree structure

**Fig 3. Tree-based pooling region representation.** The shadow area of each node indicates a pooling region, and the dotted red line of each node represents a dividing line, which divides the region of the node into two subregions.

obtained on each training image are gathered to obtain the labelled coordinate data $\boldsymbol{B}^k = \{(\boldsymbol{s}_i^j, c_j), i = 1, 2, ..., N_j, j = 1, 2, ..., M\}$ of the $k$th word, where $M$ is the number of the training images.

Each word has its own labelled coordinate data. Furthermore, the obseration windows of different sizes and steps will generate different responses to same word. Hence, if different kinds of observation windows are used, for each kind of window, each word will have the labelled coordinate data obtained with this kind of window. Providing there are $O$ kinds of observation windows, $\boldsymbol{B}^{k, 1}, \boldsymbol{B}^{k, 2}, ...\boldsymbol{B}^{k, O}$ corresponding to the $O$ kinds of windows will be generated for the $k$th word. Each word will have $O$ labelled coordinate data.

Due to the variety of image sizes, the center coordinate of image patch is normalized according to image size, in order to attain the normalized coordinate of response. In this paper, the image center is defined as the origin of coordinate system. The center coordinate $(x, y)$ of image patch represents that it is $x$ pixels and $y$ pixels away from the image center in horizontal and vertical directions, respectively. For an image of the size $a \times b$, the normalized coordinate is calculated as $(2x/a, 2y/b)$. After normalization, the horizontal and vertical coordinate values of image patch are both limited to the values from -1/2 to 1/2.

**3.3.2 Pooling region learning.** Pooling region learning of a visual word is achieved by constructing a tree for the word. The tree is constructed from the root node to the leaf nodes. The whole image region is taken as the pooling region of the root node, and the pooling region of parent node is divided into two subregions by a dividing line. In order to obtain the subregions with high discriminability, the best dividing line is found according to information gain. In detail, when dividing the pooling region of a parent node (splitting a parent node), we first learn a dividing direction by applying LDA on the labelled coordinate data from the region, and then find the best dividing line in this direction, by which the weighted entropy of the subregions of the parent region is smallest. The step of splitting node (dividing pooling region) is recursively performed until any one of the following three conditions is satisfied: (1) for a node, if its depth in the tree is larger than a user-specified threshold (maximum depth), then stop splitting the node; (2) for a node, if the number of the labelled coordinate data from its pooling region is less than a threshold value decided by user, then stop splitting the node. (3) for a node, if the labels of the labelled coordinate data from its pooling region are identical, then stop splitting the node.

The complete tree construction process is illustrated as follows:

*function* **Tree-Construction (Node *Q*)**

***step 1***: If the node $Q$ does not meet any one of the stop conditions of splitting node, then

   ***step 2***: Learn a dividing direction $p = (p_x, p_y)$ of $Q$ using the labelled coordinate data from the pooling region of $Q$, as stated below.

   ***step 3***: Learn the best dividing line $y = (p_y/p_x)x + b$ of $Q$ using the labelled coordinate data from the pooling region of $Q$, as stated below.

   ***step 4***: Create two child nodes $Q_l$ and $Q_r$ of $Q$ by the best dividing line. The data of $Q_l$ is the labelled coordinate data from the left region of the dividing line, and the data of $Q_r$ is the rest obtained by subtracting the data of $Q_l$ from the data of $Q$. The depths of $Q_l$ and $Q_r$ are both set to $d + 1$, where $d$ is the depth of $Q$.

   ***step 5***: Tree-Construction (Node $Q_l$)

   ***step 6***: Tree-Construction (Node $Q_r$)

***step 7***: End

The above function is performed recursively after inputting the root node. For the $k$th word, the data of the root node is the labelled coordinate data $\boldsymbol{B}^k$. In the following, we present in detail the method for learning the dividing direction and the method for learning the best dividing line.

**Learning of dividing direction.** Each element in image representation vector corresponds to a pooling region of visual word. Hence, the discriminability of pooling region has close relationship with the discriminability of image representation vector. In our work, the discriminability of a pooling region is evaluated by the entropy of the label distribution of the labelled coordinate data from the region. On account of that the spatial distributions of the coordinate data with different labels are not same, the weighted entropies of the subregions divided by the dividing lines of different locations and directions, are also different. To obtain the pooling subregions with high discriminability, LDA is employed to learn a dividing direction, and the best dividing line is selected from the set of the candidate dividing lines in this direction according to information gain.

The advantages of using LDA are twofold. One is that, this practice reduces the computational costs on finding the best dividing, since it avoids to build the candidate split points along each dimension and calculate the information gain for each split point. Another is that, if the coordinate distribution of each class is Gaussian-like one, compared with the set of the split points, it is likely to find the dividing with higher information gain from the set of the candidate dividing lines.

Let $B = \{(s_i, c_i), i = 1, \ldots, N\}$ be the labelled coordinate data with $C$ classes $\{\omega_c\}_{c=1}^{C}$ from a parent region, where $s_i \in \mathbb{R}^{2 \times 1}$ denotes the $i$-th coordinate datum and $c_i$ is the class label of $s_i$, the fisher criterion is defined as follows:

$$J(\boldsymbol{w}) = \frac{\boldsymbol{w}^T S_B \boldsymbol{w}}{\boldsymbol{w}^T S_W \boldsymbol{w}} \tag{1}$$

where,

$$S_B = \sum_{c=1}^{C} N_c (\boldsymbol{u}_c - \boldsymbol{u})(\boldsymbol{u}_c - \boldsymbol{u})^T$$

$$S_W = \sum_{c=1}^{C} \sum_{s_i \in \omega_c} (\boldsymbol{u}_c - \boldsymbol{u})(\boldsymbol{u}_c - \boldsymbol{u})^T \tag{2}$$

$$\boldsymbol{u}_c = \frac{1}{N_c} \sum_{s_i \in \omega_c} \boldsymbol{s}_i, \boldsymbol{u} = \frac{1}{N} \sum_{s_i \in \omega_c} N_c \boldsymbol{s}_i$$

$N_c$ is the number of the coordinate with the label $c$. Here, the learning objective is to maximize the Fisher criterion $J(\boldsymbol{w})$ under the condition $\boldsymbol{w}^T S_W \boldsymbol{w} = 1$. The objective is achieved by solving the generalized eigenvalue problem $S_B \boldsymbol{w} = \lambda S_W \boldsymbol{w}$. We retain the eigenvector $\boldsymbol{w}$ with the largest eigenvalue $\lambda_{max}$. The dividing direction $\boldsymbol{p} = (-w_y, w_x)$ is obtained by calculating the orthogonal unit vector of the eigenvector $\boldsymbol{w} = (w_x, w_y)$.

**Learning of the best dividing line.** After obtaining the dividing direction of a pooling region, we find the best dividing line from the set of the candidate ones in this direction. The weighted entropy of its subregions obtained with each candidate line is calculated, and the candidate one that corresponds to the smallest weighted entropy is selected as the best dividing line.

Given the dividing direction $\boldsymbol{p} = (p_x, p_y)$ obtained for the region $R$ and a dividing line $y = kx + b$ in this direction $\boldsymbol{p}$, where $k = p_y/p_x$ and $b$ is a vertical intercept, the weighted entropy $H$ of the entropies $H_l$, $H_r$ of the two subregions $R_l$, $R_r$ are computed as:

$$H = \frac{N_l}{N} \times H_l + \frac{N_r}{N} \times H_r, \tag{3}$$

where $R_l$ ($R_r$) is the left (right) region of this line, $N_l$ ($N_r$) is the number of the labelled coordinate data from the subregion $R_l$ ($R_r$), and $N = N_l + N_r$. The entropy $H_l$ ($H_r$) is computed on the label distribution of the labelled coordinate data from the subregion $R_l$ ($R_r$). To judge which of the two subregions a coordinate belongs to, the function $f(x, y) = y - kx - b$ is built and used as follows:

$$if k \geq 0, \begin{cases} R_l, iff(x, y) \geq 0 \\ R_r, iff(x, y) < 0 \end{cases}, if k < 0, \begin{cases} R_l, iff(x, y) \leq 0 \\ R_r, iff(x, y) > 0 \end{cases} \tag{4}$$

For example, given a coordinate $s_i = (x_i, y_i)$, if $k >= 0$ and $f(x_i, y_i) > 0$, then it belongs to the subregion $R_l$.

The best dividing line is selected from the set $S$ of the candidate dividing lines. The only difference among the candidate dividing lines is the vertical intercept $b$. Given the labelled coordinate data $\boldsymbol{B} = \{(s_i, c_i), i = 1, \ldots, N\}$ from the region $R$ and the dividing direction $\boldsymbol{p}$, we build the set $S$ by the following steps. First, we find all the vertical intercepts $I = \{b_1, b_2, \ldots, b_N\}$ by solving the formula $b = y - (p_y/p_x)x$ using each coordinate in $\boldsymbol{B}$. For the coordinate $s_i = (x_i, y_i)$, its vertical intercept $b_i$ is $y_i - (p_y/p_x)x_i$. Then, we sort $I$ in ascending order to obtain a sorted set $J = \{d_1, d_2, \ldots, d_N\}$. At last, the dividing direction $\boldsymbol{p}$ and an average value $(d_i + d_{i+1})/2$ form a candidate dividing line $y = (p_y/p_x)x + (d_i + d_{i+1})/2$. Since there are $N$ intercepts in $J$, the cardinality of the obtained set $S$ is $N - 1$. For each line in $S$, we compute its weighted entropy $H$ by formula (3). The line with the smallest entropy $H_{min}$ (maximum information gain) is selected as the best dividing line.

**3.3.3 Grouping the responses of observation window.** The responses of observation window to each word are grouped by the pooling regions of the word. The responses belonging to same pooling region form a group. It is required to know which pooling regions each response belongs to. Specifically, for a response, deciding which of the child nodes it belongs to is performed from the root node to the leaf nodes according to its coordinate and the best dividing line of non-leaf node. The pooling regions of the nodes on the decision path are the regions the response belongs to. Given the coordinate $s_i = (x_i, y_i)$ of a response and a non-leaf node $q$, as done in formula (4), the sign of $f(x_i, y_i) = y_i - k_q x_i - b_q$ decides which of its child nodes the response belongs to, where $k_q, b_q$ are the parameters about the best dividing line of the node $q$.

In fact, it is feasible that the responses are grouped only by a part of all the pooling regions of visual word, e.g., the pooling regions indicated by the nodes at the levels 0, 2, 4. This practice not only decreases the dimensionality of image representation vector, but also reduces the redundant information incurred by too many pooling regions (demonstrated in Table 4).

## 3.4 Image representation vector

The image representation vector consists of the maximums from the groups of all the visual words. If only one kind of observation window is used, each word only has one tree. For the $k$th word, the maximums $v_1^k, v_2^k, \ldots, v_G^k$ from its $G$ groups obtained with its tree are concatenated to form the vector $\boldsymbol{v}^k = (v_1^k, v_2^k, \ldots, v_G^k)^T$ of the word. The vectors $\boldsymbol{v}^1, \boldsymbol{v}^2, \ldots, \boldsymbol{v}^K$ ($K$ is the number of visual words) of all the words are jointed to form the vector $\boldsymbol{v} = ((\boldsymbol{v}^1)^T, (\boldsymbol{v}^2)^T, \ldots, (\boldsymbol{v}^K)^T)^T$. If $O$ ($O > 1$) kinds of observation windows are used, the vectors $\boldsymbol{v}_1, \boldsymbol{v}_2, \ldots, \boldsymbol{v}_O$ for all the kinds of observation windows are jointed to form the vector $\boldsymbol{v} = (\boldsymbol{v}_1^T, \boldsymbol{v}_2^T, \ldots, \boldsymbol{v}_O^T)^T$. The vector $\boldsymbol{v}$ is normalized as the image representation vector $\boldsymbol{v}_I = \|\boldsymbol{v}\|_2$ by $l_2$-normalization.

## 4 Experiments

### 4.1 Datasets

In our experiments, four small datasets are used to evaluate the classification performance of our proposed method.

*Scene-15*: It consists of 4492 images from the fifteen classes, such as bedroom, industrial, forest and so on. The number of images per class varies from 260 to 440. We chose randomly 100 images from each class to form the training set, and the remaining images are used as a test set.

*Caltech-101*: Caltech-101 dataset is a challenging object recognition dataset, which contains 9,144 images in 101 object classes and one background class. The number of images per class ranges from 31 to 800. We consider 30 training images and up to 30 testing images per class.

*Caltech-256*: Caltech-256 dataset consists of 257 object classes. There are 30607 images in total. Compared with Caltech-101, it presents much higher variability in object size, location and poses. 30 images and 20 images from each class are used for training and testing, respectively.

*Corel-10*: It contains 1,000 images in 10 classes (flower, elephant, owls, tiger, building, beach, skiing, horses, mountains, food). For evaluation, the images were randomly divided into 50 training and 50 test images for each class.

## 4.2 Implementation details

For images from all the datasets, we extract the dense patches of $16 \times 16$ pixels. The step between two neighboring patches are set to 8 pixels for Scene-15 and Corel-10, 6 pixels for Caltech-101 and Caltech-256. Each patch is described as a SIFT descriptor (128-dimensional vector). We use the *K*-means implemented by VLFeat [36] to learn visual dictionary. The dictionary size is set to 1024 for Scene-15 and Corel-10, and 2048 for Caltech-101 and Caltech-256. Localized Soft-assignment Coding (LSaC) [37] is applied to encode the SIFT descriptors using the learned dictionary owing to its superior performance to sparse coding [38] and Locality-constrained Linear Coding [39]. As suggested in [36], the number of visual words to encode a descriptor is set to 5. In this case, in the place where a descriptor locates at, five different visual words appear simultaneously. For all the datasets, a one-versus-rest linear SVM for each class is trained. We adopt [37] as our baseline, termed as *SPM(baseline)*. All the experiments are conducted on a 64-bit Windows 10 with Intel Core i5-4590 at 3.30 GHz * 4 on 16GB RAM.

In order to evaluate if the classification accuracy is improved after applying our method more accurately, the following experimental setups are taken. For each dataset, we only randomly select the training images and testing images 10 times to obtain 10 fixed training sets and testing sets. 10 fixed dictionaries are learned on the fixed training sets respectively. The coding and pooling strategies adopted by our method and *SPM(baseline)* are also same. In this case, the only factor that influences the classification accuracy, is pooling region. For each experiment setup about pooling region learning, we conduct the experiment 10 times on the 10 fixed training and testing sets, and report the average of the classification accuracies of 10 experiments. The average classification accuracy of *SPM(baseline)* is also obtained on the 10 fixed training and testing sets.

## 4.3 Impact of the nodes at different levels

The experiments are started with an in-depth analysis of the discriminability of the pooling regions indicated by the nodes at different levels. We investigate the nodes at the level 0 to 6, respectively. For the level *l*, the number of the nodes at this level is $2^l$. In SPM, the blocks obtained by dividing the whole image region at the resoluation levels of $1 \times 1$, $2 \times 2$, $4 \times 4$ and $8 \times 8$, correspond to the pooling regions of the nodes at the level 0, 2, 4 and 6. Here, we only use one kind of observation window. Its size and step are set to $1 \times 1$ and 1.

Table 1 shows the classification accuracies corresponding to the different levels. For all the datasets, the accuracies obtained by our method are superior to the baseline consistently. The results demonstrate that the pooling regions learned are more discriminative than the ones of SPM. We note the drop of the accuracy after some level, for example, the level 4 for Scene-15 and the level 5 for Caltech-101. This can be explained by the noise incurred by too finer

**Table 1. Results for different levels on Scene-15, Caltech-101, Caltech-256 and Corel-10.** SPM(base.) is the abbreviation for SPM(baseline).

| Dataset | Method | Level | | | | | | |
|---|---|---|---|---|---|---|---|---|
| | | **0** | **1** | **2** | **3** | **4** | **5** | **6** |
| Scene-15 | Our | 71.08(0.28) | 80.58(0.25) | **82.62(0.20)** | 83.36(0.42) | **82.97(0.57)** | 82.77(0.42) | **82.67(0.11)** |
| | SPM(base.) | | - | 79.92(0.14) | - | 81.08(0.40) | - | 78.91(0.10) |
| Caltech-101 | Our | 52.20(0.39) | 65.34(0.85) | **69.50(1.18)** | 70.11(1.34) | **72.86(0.54)** | 74.06(0.90) | **73.24(1.15)** |
| | SPM(base.) | | - | 67.55(1.19) | - | 72.04(0.57) | - | 69.17(0.41) |
| Caltech-256 | Our | 24.87(0.71) | 32.09(0.79) | **35.61(0.57)** | 37.14(0.75) | **37.15(0.40)** | 36.39(0.56) | **35.63(0.56)** |
| | SPM(base.) | | - | 31.59(1.39) | - | 34.46(0.33) | - | 31.60(0.46) |
| Corel-10 | Our | 83.93(0.75) | 87.00(0.95) | **88.07(0.68)** | 87.27(0.64) | **87.13(0.51)** | 86.47(0.81) | **85.47(0.80)** |
| | SPM(base.) | | - | 86.33(0.80) | - | 85.60(0.27) | - | 84.20(0.82) |

dividing. Besides, we can achieve the similar accuracies to the best obtained by SPM with lower dimensionality. For Scene-15, the accuracy 82.62% (4096 dimensions) of the single level 2 is very close to 82.84% (21504 dimensions, shown in Table 5). For Caltech 101, Caltech 256 and Corel-10, we can draw the same conclusion (shown in Tables 6–8). We also note that, for all the datasets, the accuracies of the level 6 are higher than SPM obviously. This means that, compared with SPM, the pooling regions learned by our method are more robust to the noise incurred by too finer dividing.

## 4.4 Impact of observation window

In this section, we investigate the impact of the size and step of observation window. A series of parameter combinations about size and step are tested. Here, the nodes at the levels 0, 2 and 4 are selected for grouping the responses.

Tables 2 and 3 show the accuracies obtained on Scenes-15 and Caltech-101, respectively. From these tables, we find that, the size and step of observation window can influence the classification accuracy. Fig 4 shows the accuracies in terms of the area of observation window.

**Table 2. Results for different kinds of observation windows on Scene-15.**

| | | horizontal size (horizontal step) | | | | |
|---|---|---|---|---|---|---|
| | | **1(1)** | **2(1)** | **3(2)** | **4(2)** | **5(3)** |
| *vertical size (vertical step)* | 1(1) | 84.25(0.38) | 84.07(0.63) | 84.32(0.09) | 84.48(0.31) | 84.40(0.36) |
| | 2(1) | 84.51(0.26) | 84.38(0.27) | 84.54(0.49) | 84.44(0.34) | 84.22(0.22) |
| | 3(2) | 84.72(0.52) | 84.43(0.43) | 84.56(0.64) | 84.64(0.55) | 84.49(0.35) |
| | 4(2) | 84.68(0.25) | 84.41(0.24) | 84.54(0.21) | 84.82(0.62) | 84.40(0.40) |
| | 5(3) | 84.49(0.10) | 84.51(0.16) | 84.67(0.28) | 84.64(0.52) | **84.89(0.31)** |

**Table 3. Results for different kinds of observation windows on Caltech-101.**

| | | horizontal size (horizontal step) | | | | |
|---|---|---|---|---|---|---|
| | | **1(1)** | **2(1)** | **3(2)** | **4(2)** | **5(3)** |
| *vertical size (vertical step)* | 1(1) | 73.93(0.84) | 73.93(0.55) | 74.02(0.25) | 74.31(0.46) | 74.46(0.56) |
| | 2(1) | 74.13(0.56) | 74.15(0.74) | 74.50(0.51) | 74.01(0.67) | 74.31(0.51) |
| | 3(2) | 73.96(0.51) | 74.26(0.38) | 74.46(0.39) | 74.42(0.47) | 74.60(0.53) |
| | 4(2) | 74.12(0.81) | 74.20(0.42) | 74.26(0.51) | 74.59(0.33) | 74.53(0.28) |
| | 5(3) | 74.50(0.45) | 74.51(0.47) | **74.68(0.35)** | 74.68(0.39) | 74.56(0.67) |

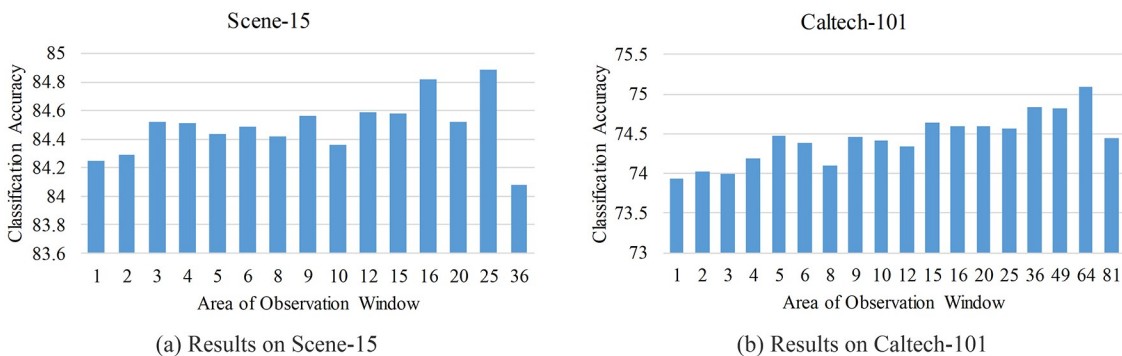

**Fig 4. Results for the observation windows of different areas on Scene-15 and Caltech-101.** The area of the window of the size $w \times h$ is calculated by $w \times h$. For the windows of same area, such as $2 \times 5$ and $5 \times 2$, the mean accuracy of these windows is reported. ($36 = 6(3) \times 6(3)$, $49 = 7(4) \times 7(4)$, $64 = 8(4) \times 8(4)$, $81 = 9(5) \times 9(5)$).

Overall, the slightly larger window can achieve higher accuracy. For Scene-15, the highest accuracy 84.89% is obtained when the size of observation window is set to $5 \times 5$, and for Caltech-101, the window of the size $8 \times 8$ leads to the highest accuracy 75.09%. However, an obvious drop appears when the size of window exceeds some size, such as $5 \times 5$ for Scene-15 and $8 \times 8$ for Caltech-101. The reason is that, the robust of image representation to image variability is improved by checking the existence of word in slightly larger window, but the information on the spatial distribution of word will loss when applying the larger window like $6 \times 6$. Moreover, we also note that, for Scene-15, when the aspect ratio $r$ of observation window is less than 1, the accuracies are higher than the ones obtained when $r > 1$, such as 84.72% for $1 \times 3$ vs. 84.32% for $3 \times 1$. Nevertheless, we cannot observe the same phenomenon on Caltech-101. Compared to the accuracies shown in Table 1, the accuracies obtained with the nodes at the levels 0, 2 and 4 are higher.

## 4.5 Impact of the combination of observation window and level of tree

We evaluate the classification accuracy of the combination of observation window and level of tree. Table 4 presents the results obtained with some common combinations. The combinations corresponding to the highest accuracies are quite different, which are (F, L2), (B, L2), (B, L3) and (A, L1). For Scene-15, Caltech-101 and Caltech-256, L2 and L3 both lead to better results than L1 with higher dimensionality. The accuracies obtained with the observation window B are higher than with the window A on the three datasets. The reason is that the slightly larger window improves the robust of image representation to image variability. Besides, for Scenes-15 and Caltech-256, the accuracy can be further improved by applying different kinds of observation windows simultaneously, while this phenomenon does not appear on Caltech-101. For Corel-10, L2 and L3 do not boost the classification accuracy, and the larger window B also does not work. The reason is that, for most of the classes in Corel-10, the spatial distributions of the features relevant highly to image class have obvious difference for the images from same class.

## 4.6 Accuracy comparison

In this section, we compare our method with **SPM(baseline)** under the same experimental setup (illustrated in Section 4.2). In addition, we also list the results reported by some representative methods, including the deep learning methods [40, 41, 44, 45, 50] and the BoVW

**Table 4. Results for some common combinations on different datasets.** (A: $1(1) \times 1(1)$; B: $4(2) \times 4(2)$; C: $4(2) \times 1(1)$; D: $1(1) \times 4(2)$; E: A, B; F: B, C, D; L1: levels 0, 2, 4; L2: levels 1, 3, 5; L3: levels 0, 1, 2, 3, 4, 5).

|  | OW | L1 | L2 | L3 |
|---|---|---|---|---|
| Scene-15 | A | 84.25(0.38) | 84.28(0.30) | 84.41(0.27) |
|  | B | 84.82(0.62) | 84.68(0.33) | 84.87(0.42) |
|  | E | 84.69(0.06) | 85.10(0.11) | 85.01(0.26) |
|  | F | 84.95(0.05) | **85.24(0.42)** | 85.16(0.13) |
| Caltech-101 | A | 73.93(0.84) | 74.67(1.39) | 74.71(0.87) |
|  | B | 74.59(0.33) | **75.83(1.15)** | 75.18(0.59) |
|  | E | 74.49(0.82) | 75.38(0.55) | 74.82(0.49) |
|  | F | 74.32(0.42) | 75.04(0.39) | 74.75(0.45) |
| Caltech-256 | A | 38.64(0.55) | 38.65(1.06) | 39.12(0.65) |
|  | B | 39.22(0.59) | 39.70(0.81) | **39.95(0.62)** |
|  | E | 39.38(0.58) | 39.50(0.51) | 39.51(0.50) |
|  | F | 39.66(0.45) | 39.81(0.41) | 39.91(0.62) |
| Corel-10 | A | **88.92(0.61)** | 88.20(0.69) | 88.44(0.43) |
|  | B | 88.32(0.33) | 87.88(0.54) | 88.08(0.30) |
|  | E | 88.60(0.51) | 88.36(0.30) | 88.20(0.49) |
|  | F | 88.64(0.56) | 88.40(0.35) | 88.28(0.52) |

methods [38, 42, 48]. The results of these methods are obtained without the help of transfer learning, i.e., using a pre-learned CNN to extract image features.

As shown in Tables 5–8, compared with **SPM(baseline)**, our method improves classification accuracy by about 1% to 2.5%. This phenomenon demonstrates that, the practice of

**Table 5. Classification accuracy on Scene-15.**

| Method | Accuracy, % |
|---|---|
| Our(F, L2) | 85.24(0.42) |
| SPM(baseline) | 82.84(0.30) |
| CDBN [40] | 78.52(0.63) |
| SS-RBM [41] | 84.1(0.8) |
| Y. Boureau et al. [42] | 83.1(0.7) |
| Y. Boureau et al. [2] | 85.6(0.2) |
| H. Goh et al. [43] | 85.2(0.5) |

**Table 6. Classification accuracy on Caltech-101.**

| Method | Accuracy, % |
|---|---|
| Our(B, L2) | 75.83(1.15) |
| SPM(baseline) | 73.81(0.42) |
| CDBN [40] | 65.4(0.5) |
| SS-RBM [41] | 75.1(1.2) |
| CNN [44] | 66.3(1.5) |
| Deconvolutional Network [45] | 66.9(1.1) |
| Hierarchical SC [46] | 74.0(1.5) |
| NBNN kernel [47] | 75.2(1.2) |
| ScSPM [38] | 73.2(0.5) |
| SCDAE [48] | 78.6(1.2) |

**Table 7. Classification accuracy on Caltech-256.**

| Method | Accuracy, % |
|---|---|
| Our(B, L3) | 39.95(0.62) |
| SPM(baseline) | 37.48(0.68) |
| Graph-matching kernel [49] | 38.1(0.6) |
| CRBM [50] | 42.1 |
| H. Goh et al. [43] | 41.5(0.7) |

**Table 8. Classification accuracy on Corel-10.**

| Method | Accuracy, % |
|---|---|
| Our(A, L1) | 88.92(0.61) |
| SPM(baseline) | 88.08(0.72) |
| ScSPM [38] | 86.6(1.01) |
| LSC [37] | 88.76(0.76) |

making each word have its own pooling regions is beneficial to image classification task. However, our results are lower in comparison with some methods [2, 43, 48, 50]. Although the accuracies obtained by our method are not attractive enough, our method is easy to be combined with a number of BoVW methods to achieve higher classification accuracy (illustrated in Section 4.7). We also note that, the deep learning methods do not achieve obvious improvement over the BoVW methods due to the lack of training data.

We compute the average computational time for Caltech-101 spent on converting an image to an image representation vector. The average time (0.76s) required by our method is slightly more than the average time (0.61s) of SPM(baseline). The time spent on pooling region learning is about 380s when the observation window of $4(2) \times 4(2)$ is used.

## 4.7 Discussion

The advantages and disadvantages of our method are shown as below:

- Our method improves the discriminability of image representation vector by learning its pooling regions for each word. However, there are computational costs associated with pooling region learning. For Caltech 101, the time spent on pooling region learning is about 380s when the observation window of $4(2) \times 4(2)$ is used. Besides, the time spent on converting an image to an image representation vector increases slightly.

- Although the classification accuracies obtained by our method are not attractive enough compared with some existing methods [2, 43, 48, 50], our method is easy to be combined with a number of BoVW methods to achieve higher classification accuracy. The reason is that our method only involves in the stage of feature pooling. The existing works focusing on feature extraction, feature description, dictionary learning and feature coding can be used in conjunction with our method. Besides, the works on Analysis Dictionary Learning (ADL) can also be applied on the image representation vector obtained by our method, resulting in more discriminative image representation vector.

- The effectiveness of our method depends on inter and intra-class visual diversity to a great extent. When the spatial distributions of the local features relevant highly to image class are similar for the images from same class, our method works well. For example, the accuracy

improvement of about 2% to 2.5% is obtained on Scene-15, Caltech-101 and Caltech-256. For most of the classes in the three datasets, the images are aligned artificially well. By contrast, only the gain of 0.9% is acquired on Corel-10 due to the large intra-class visual diversity.

## 5 Conclusion

In this paper, we proposed to make each word have its own pooling regions, and raised a simple yet effective method for learning pooling region. A kind of small window named observation window was proposed to obtain its responses to each word over the whole image region. The pooling regions of visual word were learned by constructing a tree with the coordinates of responses and image class labels. The classification accuracy was improved by about 1% to 2.5% after applying our method. This phenomenon demonstrates that the practice of making each word have its own pooling regions is beneficial to image classification task. Furthermore, we found by our method that, the image representation vector obtained with a slightly larger observation window (e.g. $4 \times 4$) achieves higher accuracy than with a small window (e.g., $1 \times 1$). The classification accuracy is likely to be improved further by applying the observation windows of different sizes and steps simultaneously. The future works we are pursuing are: 1) learning the pooling regions of visual phrase; 2) applying the thought of our method on the convolution layers of CNN.

## Author Contributions

**Conceptualization:** Ye Xu, Zezhong Xu.

**Data curation:** Ye Xu, Xiaodong Yu, Tian Wang.

**Formal analysis:** Ye Xu, Xiaodong Yu, Tian Wang.

**Investigation:** Xiaodong Yu.

**Methodology:** Ye Xu.

**Project administration:** Zezhong Xu.

**Resources:** Tian Wang, Zezhong Xu.

**Software:** Ye Xu, Tian Wang.

**Supervision:** Xiaodong Yu.

**Validation:** Ye Xu, Tian Wang.

**Visualization:** Ye Xu, Tian Wang.

**Writing – original draft:** Ye Xu, Xiaodong Yu.

**Writing – review & editing:** Ye Xu, Xiaodong Yu, Zezhong Xu.

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
