## [Decision Letter · Decision Letter 0]

25 Feb 2020

PONE-D-20-01860

Pooling Region Learning of Visual Word for Image Classification using Bag-of-Visual-Words Model

PLOS ONE

Dear Dr. Xu,

Thank you for submitting your manuscript to PLOS ONE. After careful consideration, we feel that it has merit but does not fully meet PLOS ONE’s publication criteria as it currently stands. Therefore, we invite you to submit a revised version of the manuscript that addresses the points raised during the review process.

Please correct the manuscript according to the comments of all reviewers. Please reply to all reviewers' comments point by point.

We would appreciate receiving your revised manuscript by Apr 10 2020 11:59PM. To enhance the reproducibility of your results, we recommend that if applicable you deposit your laboratory protocols in protocols.io, where a protocol can be assigned its own identifier (DOI) such that it can be cited independently in the future. For instructions see: http://journals.plos.org/plosone/s/submission-guidelines#loc-laboratory-protocols

We look forward to receiving your revised manuscript.

Kind regards,

Paweł Pławiak, Ph.D.

Academic Editor

PLOS ONE

Additional Editor Comments (if provided):

Please correct the manuscript according to the comments of all reviewers. Please reply to all reviewers' comments point by point.

Journal Requirements:

Reviewers' comments:

Reviewer's Responses to Questions

**Comments to the Author**

1. Is the manuscript technically sound, and do the data support the conclusions?

Reviewer #1: Partly

Reviewer #2: Yes

Reviewer #3: Yes

2. Has the statistical analysis been performed appropriately and rigorously? 

Reviewer #1: Yes

Reviewer #2: Yes

Reviewer #3: Yes

3. Have the authors made all data underlying the findings in their manuscript fully available?

Reviewer #1: Yes

Reviewer #2: No

Reviewer #3: Yes

4. Is the manuscript presented in an intelligible fashion and written in standard English?

Reviewer #1: Yes

Reviewer #2: Yes

Reviewer #3: Yes

5. Review Comments to the Author

Reviewer #1: • The abstract can be rewritten to be more meaningful. The authors should add more details about their final results in the abstract. Abstract should clarify what is exactly proposed (the technical contribution) and how the proposed approach is validated.

• The structure of the paper should be added to the end of Introduction Section.

• The paper does not explain clearly its advantages with respect to the literature: it is not clear what is the novelty and contributions of the proposed work: does it propose a new method? Or does the novelty only consist in the application?

• Do the authors employ any cross-validation scheme? Please, provide details about it.

• Please highlight the advantages and disadvantages of your method.

• Please add more future works in Conclusion Section.

Reviewer #2: With their work, the authors address the problem of image classification using Bag-of-Visual-Words model. Their contributions fall in proposing a new pooling method which is through learning pooling regions for each word. This learning is occurred by adopting a tree structure and then the pooling regions are achieved.

The method is clearly discussed and the results are good.

Here are some questions regarding the proposed method and results and some suggestions to improve the quality of the paper:

- The authors claim that BoVW is better than the existing methods like CNN in terms of memory and time, but there is no evidence in the results that proves this claim. At least in terms of time, there should be some comparison between these two approaches. My other concern is that using Deep CNN models running on a GPU may result in more accurate results in appropriate time.

- Tables 5 to 8 show the experimental results for different data sets and for different classification algorithms including the authors’ work. Each table compares a set of different methods. So I think it is hard to conclude that the proposed approach achieves good results compared to other works for a fair number of data sets. How the authors justify these findings?

- In their work, the authors define the best dividing direction as the projection direction of LDA method. Why is this a good choice?

- While the visual presentation of the paper consists of comprehensible figures and is acceptable, the steps of the proposed method has not well described and don't precisely show the aim of the paper. It is required that authors show the major steps of the whole classification task in order, including the proposed tree structure model and construction of pooling regions, and not just tree construction part of the work.

In general, the writing of the paper is good and the proposed idea is interesting, but the algorithm should be explained better and the results are needed to be improved.

Reviewer #3: In this work, the authors try to classify small image datasets using the bag of visual words instead of CNN. In this process, the authors learn some pooling regions for each visual word to capture the spatial distribution of the local features. Multiple training data was learned using different window selections to make classification better. The authors have performed rigorous experimental conditions. The paper is written in a meaningful manner. All the mathematical and experimental details are adequately articulated.

My only concern is, please providing any inherent limitations and future directions of the proposed method.

6. PLOS authors have the option to publish the peer review history of their article (what does this mean?). If published, this will include your full peer review and any attached files.

Reviewer #1: No

Reviewer #2: No

Reviewer #3: No

---

## [Author Response · Author response to Decision Letter 0]

23 Apr 2020

Dear Dr. Plawiak:

Subject: Submission of revised paper PONE-D-20-01860

Thank you for your email dated 26 February 2020 enclosing the reviewer’s comments. We have carefully reviewed the comments and have revised the manuscript accordingly. Our responses are given in a point-by-point manner below.

 We hope the revised version is now suitable for publication and look forward to hearing from you in due course.

 Sincerely,

 Ye Xu, Ph. D

 Wuxi Institute of Technology 

Response to Reviewer 1: Thank you for your review of our paper. We reorganized the content of our paper and added more illustrations about our proposed method. Some inaccurate descriptions were also modified carefully. Besides, we also added new content including the process of image representation, the advantages and disadvantages and the future work. 

The major changes are shown as follows: 

1. We added a new section (Section 3.1) named “Process of Image Representation”. In this section, we explained the five stages of the BoVW model in order, and illustrated which stage our method involves in. The key steps of our work were also given in order. A new figure (Figure 1) was presented in the revised manuscript, which shows the process of image representation including our method. 

2. We added a new section (Section 3.2) named “Observation Window”, which corresponds to the first step of our method. In this section, we introduced the principle of observation window. The advantages of using observation window were also stated. 

3. We added a new section (Section 3.3) named “Pooling Regions of Visual Word”, which corresponds to the second step of our method. At the beginning of this section, we illustrated the reason why the practice of making each word have its own pooling regions is feasible. Afterwards, we described how to learn its pooling region for each word. The learning process was divided into three small steps, which are illustrated in Section 3.3.1, Section 3.3.2 and Section 3.3.3, respectively. 

a) In Section 3.3.1 (Labelled Coordinate Data), we explained the reason why our method needs to build labelled coordinate data. Then, the details on building data were presented. 

b) In Section 3.3.2 (Pooling Region Learning), we first gave the entire learning process of pooling regions. Two key problems in the learning process are introduced in the subsection “Learning of Dividing Direction” and the subsection “Learning of the Best Dividing Line”. 

i. In the subsection “Learning of Dividing Direction”, we explained why LDA is used to learn a dividing direction. The details on how to learn dividing direction were also presented. 

ii. In the subsection “Learning of the Best Dividing Line”, the learning process of the best dividing Line was illustrated in detail. 

c) In Section 3.3.3 (Grouping the Responses of Observation Window), we illustrated how to group the responses of observation window by pooling regions. The advantage of using a part of all the pooing regions was also given. 

4. We added a new section (Section 3.4) named “Image Representation Vector”, which corresponds to the third step of our method. In this section, we explained how to obtain the image representation vector with the maximums from the groups of all the visual words. 

5. We added a new section (Section 4.7) named “Discussion”. In this section, the advantages and disadvantages of our method were illustrated. 

6. We revised the abstract, the introduction and the conclusion. In Abstract Section, we added more details about how to validate our method, and our results. In Introduction Section, we modified the description about our method. In Conclusion Section, we added the future works of our method. 

7. We added more details about our experiment setup in Section 4.2 (Implementation details).

8. Section 3 of the original manuscript was removed. 

Reviewer #1: 

1. The abstract can be rewritten to be more meaningful. The authors should add more details about their final results in the abstract. Abstract should clarify what is exactly proposed (the technical contribution) and how the proposed approach is validated. 

Answer: “In the problem where there is not enough data to use Deep Learning, Bag-of-Visual-Words (BoVW) is still a good alternative for image classification. In BoVW model, many pooling methods are proposed to incorporate the spatial information of local feature into the image representation vector, but none of the methods devote to making each visual word have its own pooling regions. The practice of designing the same pooling regions for all the words restrains the discriminability of image representation, since the spatial distributions of the local features indexed by different visual words are not same. In this paper, we propose to make each visual word have its own pooling regions, and raise a simple yet effective method for learning pooling region. Concretely, a kind of small window named observation window is used to obtain its responses to each word over the whole image region. The pooling regions of each word are organized by a kind of tree structure, in which each node indicates a pooling region. For each word, its pooling regions are learned by constructing a tree with its labelled coordinate data. The labelled coordinate data consist of the coordinates of responses and image class labels. The effectiveness of our method is validated by observing if there is an obvious classification accuracy improvement after applying our method. Our experimental results on four small datasets (i.e., Scene-15, Caltech-101, Caltech-256 and Corel-10) show that, the classification accuracy is improved by about 1% to 2.5%. We experimentally demonstrate that the practice of making each word have its own pooling regions is beneficial to image classification task, which is the significance of our work.”

2. The structure of the paper should be added to the end of Introduction Section.

Answer: We added the description about the structure of the paper as follows. “The remainder of this paper is organized as follows: the proceeding section is about the related works. Section 3 illustrates our work in detail. Experimental evaluation and analysis are reported in Section 4, and the conclusion is drawn in Section 5.”

3. The paper does not explain clearly its advantages with respect to the literature: it is not clear what is the novelty and contributions of the proposed work: does it propose a new method? Or does the novelty only consist in the application?

Answer: In this paper, we proposed to make each word have its own pooling regions. A simple yet effective method for learning its pooling regions for each word, was raised. We experimentally demonstrate that, the practice of making each word have its own pooling regions, is beneficial to image classification task. This is the main significance of our work. 

Furthermore, there are some small innovations in our proposed method. The first is that, we introduced a kind of small window named observation window to obtain the existences of visual word in the small windows placed on the whole image region (The existences of visual word are named the responses of observation window to visual word for clarity). The image representation vector was obtained based on the responses of observation window instead of the coding coefficients. The second is that, we proposed to organize the pooling regions with a kind of tree structure. The pooling regions of visual word were learned by constructing a tree. The third is that, LDA was adopted to learn a dividing direction when dividing a pooling region of parent node. 

Some new findings were also found by our proposed method: 1) the image representation vector obtained with the responses of observation window achieves higher accuracy than with the coding coefficients; 2) the classification accuracy can be improved by applying the observation window of different sizes and steps simultaneously. 

4. Do the authors employ any cross-validation scheme? Please, provide details about it.

Answer: We did not employ any cross-validation scheme in our experiments. However, our experiments followed the major of the existing works such as [13][14][18][32][37][39] and so on. As illustrated in [14], the performance measure can be biased if test set sizes for different classes vary significantly. This is especially true of Caltech-101 and Caltech-256, where some of the “easiest” classes are disproportionately large.

In our experiments, for each dataset, we only randomly selected the training images and testing images 10 times to obtain 10 fixed training sets and testing sets. 10 fixed dictionaries were learned on the fixed training sets respectively. The coding and pooling strategies adopted by our method and the baseline are also same. In this case, the only factor that influences the classification accuracy, is pooling region. For each experiment setup about pooling region learning, we conducted the experiment 10 times on the 10 fixed training and testing sets, and reported the average of the classification accuracies of 10 experiments. The average classification accuracy of the baseline was also obtained on the 10 fixed training and testing sets.

5. Please highlight the advantages and disadvantages of your method.

Answer: We added a new section named “Discussion”. In this section, the advantages and disadvantages of our method are stated as follows:

Our method improves the discriminability of image representation vector by learning its pooling regions for each word. However, there are computational costs associated with pooling region learning. For Caltech 101, the time spent on pooling region learning is about 380s when the observation window of 4(2)×4(2) is used. Besides, the time spent on converting an image to an image representation vector increases slightly.

Although the classification accuracies obtained by our method are not attractive enough compared with some existing methods [2, 43, 48, 50], our method is easy to be combined with a number of BoVW methods to achieve higher classification accuracy. The reason is that our method only involves in the stage of feature pooling. The existing works focusing on feature extraction, feature description and feature coding can be used in conjunction with our method. Besides, the works on Analysis Dictionary Learning (ADL) can also be applied on the image representation vector obtained by our method, resulting in more discriminative image representation vector.

The effectiveness of our method depends on inter and intra-class visual diversity to a great extent. When the spatial distributions of the local features relevant highly to image class are similar for the images from same class, our method works well. For example, the accuracy improvement of about 2% to 2.5% is obtained on Scene-15, Caltech-101 and Caltech-256. For most of the classes in the three datasets, the images are aligned artificially well. By contrast, only the gain of 0.9% is acquired on Corel-10 due to the large intra-class visual diversity.

6. Please add more future works in Conclusion Section.

Answer: We added the future works in Conclusion Section as follows. “The future works we are pursuing are: 1) learning the pooling regions of visual phrase; 2) applying the thought of our method on the convolution layers of CNN.”

Response to Reviewer 2: Thank you for your review of our paper. We have answered each of your points below.

Reviewer #2: 

1. The authors claim that BoVW is better than the existing methods like CNN in terms of memory and time, but there is no evidence in the results that proves this claim. At least in terms of time, there should be some comparison between these two approaches. My other concern is that using Deep CNN models running on a GPU may result in more accurate results in appropriate time.

Answer: We thank you for pointing out the inaccurate description about the comparison between BoVW and CNN. Owing to the powerful computation ability of GPU, the speed of classifying image using CNN model is faster in many cases. However, the success of CNN depends on a huge amount of data, and inevitably needs a lot of time for training. When dealing with some classification tasks that only provide a small number of data for training, BoVW might work well since it does not require any prior initialization and very time-consuming training. Moreover, BoVW model has evolved in an understandable way in the past 15 years or so. By analyzing the target classification task, it is feasible to make use of human knowledge obtained to improve BoVW model from the aspects of feature extraction, feature description, dictionary learning, feature coding and feature pooling. Therefore, for some simple tasks, BoVW model is probably capable of attaining satisfactory results. As far as we know, there are a number of BoVW methods that devoted to exploring the generalized factors relevant to image classification. In our paper, we mainly explored what would happen if each word has its own pooling regions. A simple yet effective method for pooling region learning is also presented. We experimentally demonstrate that the practice of making each word have its own pooling regions is beneficial to image classification task, which is the significance of our work.

As for the classification accuracy, although the accuracies obtained by our method are not attractive enough compared with some existing methods, our method is easy to be combined with a number of BoVW methods to achieve higher classification accuracy. The reason is that our method only involves in the stage of feature pooling. The existing works focusing on feature extraction, feature description, dictionary learning, and feature coding can be used jointly with our method. Besides, the works on Analysis Dictionary Learning (ADL) can also be applied on the image representation vector obtained by our method, resulting in more discriminative image representation. On the other hand, our work can be extended in two directions: 1) learning the pooling regions of visual phrase; 2) applying the thought of our method on the convolution layers of CNN.

We removed the inaccurate word “efficient” in the description about BoVW model in Introduction Section.

2. Tables 5 to 8 show the experimental results for different data sets and for different classification algorithms including the authors’ work. Each table compares a set of different methods. So I think it is hard to conclude that the proposed approach achieves good results compared to other works for a fair number of data sets. How the authors justify these findings?

Answer: We thank you for pointing out the inaccurate description about accuracy comparison. Indeed, the results obtained by our method are not attractive enough, but our work is easy to be combined with other methods to achieve higher classification accuracy. We revised the description about accuracy comparison as follows. “As shown in Tables. 5-8, compared with SPM(baseline), our method improves classification accuracy by about 1% to 2.5%. This phenomenon demonstrates that, the practice of making each word have its own pooling regions is beneficial to image classification task. However, our results are lower in comparison with some methods [2, 43, 48, 50]. Although the accuracies obtained by our method are not attractive enough, our method is easy to be combined with a number of BoVW methods to achieve higher classification accuracy (illustrated in Section 4.7). We also note that, the deep learning methods do not achieve obvious improvement over the BoVW methods due to the lack of training data.”

3. In their work, the authors define the best dividing direction as the projection direction of LDA method. Why is this a good choice?

Answer: In this paper, we adopted LDA to learn a dividing direction. The advantages of using LDA are twofold. One is that, this practice reduces the computational costs on finding the best dividing, since it avoids to build the candidate split points along each dimension and calculate the information gain for each split point. Another is that, if the coordinate distribution of each class is Gaussian-like one, compared with the set of the split points, it is likely to find the dividing with higher information gain from the set of the candidate dividing lines. Indeed, the word “best” is an inaccurate embellishment to the word “dividing direction”. We removed the word “best”, and revised the original section 3.2.1 (Learning of the Best Dividing Direction) as follows:

“Learning of Dividing Direction Each element in image representation vector corresponds to a pooling region of visual word. Hence, the discriminability of pooling region has close relationship with the discriminability of image representation vector. In our work, the discriminability of a pooling region is evaluated by the entropy of the label distribution of the labelled coordinate data from the region. On account of that the spatial distributions of the coordinate data with different labels are not same, the weighted entropies of the subregions divided by the dividing lines of different locations and directions, are also different. To obtain the pooling subregions with high discriminability, LDA is employed to learn a dividing direction, and the best dividing line is selected from the set of the candidate dividing lines in this direction according to information gain. 

The advantages of using LDA are twofold. One is that, this practice reduces the computational costs on finding the best dividing, since it avoids to build the candidate split points along each dimension and calculate the information gain for each split point. Another is that, if the coordinate distribution of each class is Gaussian-like one, compared with the set of the split points, it is likely to find the dividing with higher information gain from the set of the candidate dividing lines.….”

4. While the visual presentation of the paper consists of comprehensible figures and is acceptable, the steps of the proposed method has not well described and don't precisely show the aim of the paper. It is required that authors show the major steps of the whole classification task in order, including the proposed tree structure model and construction of pooling regions, and not just tree construction part of the work.

Answer: We reorganized the content of our paper and added more illustrations about our proposed method. Some inaccurate descriptions were also modified carefully. Besides, we also added new content including the process of image representation, the advantages and disadvantages and the future work. As for the experimental results, although the results are not attractive, they should be enough to support the aim of our paper in our opinion. In future, we will devote to exploring the more advanced learning method and learning the pooling regions for visual phrase, in order to obtain better results.

The major changes are shown as follows: 

1. We added a new section (Section 3.1) named “Process of Image Representation”. In this section, we explained the five stages of the BoVW model in order, and illustrated which stage our method involves in. The key steps of our work were also given in order. A new figure (Figure 1) was presented in the revised manuscript, which shows the process of image representation including our method. 

2. We added a new section (Section 3.2) named “Observation Window”, which corresponds to the first step of our method. In this section, we introduced the principle of observation window. The advantages of using observation window were also stated. 

3. We added a new section (Section 3.3) named “Pooling Regions of Visual Word”, which corresponds to the second step of our method. At the beginning of this section, we illustrated the reason why the practice of making each word have its own pooling regions is feasible. Afterwards, we described how to learn its pooling region for each word. The learning process was divided into three small steps, which are illustrated in Section 3.3.1, Section 3.3.2 and Section 3.3.3, respectively. 

a) In Section 3.3.1 (Labelled Coordinate Data), we explained the reason why our method needs to build labelled coordinate data. Then, the details on building data were presented. 

b) In Section 3.3.2 (Pooling Region Learning), we first gave the entire learning process of pooling regions. Two key problems in the learning process are introduced in the subsection “Learning of Dividing Direction” and the subsection “Learning of the Best Dividing Line”. 

i. In the subsection “Learning of Dividing Direction”, we explained why LDA is used to learn a dividing direction. The details on how to learn dividing direction were also presented. 

ii. In the subsection “Learning of the Best Dividing Line”, the learning process of the best dividing Line was illustrated in detail. 

c) In Section 3.3.3 (Grouping the Responses of Observation Window), we illustrated how to group the responses of observation window by pooling regions. The advantage of using a part of all the pooing regions was also given. 

4. We added a new section (Section 3.4) named “Image Representation Vector”, which corresponds to the third step of our method. In this section, we explained how to obtain the image representation vector with the maximums from the groups of all the visual words. 

5. We added a new section (Section 4.7) named “Discussion”. In this section, the advantages and disadvantages of our method were illustrated. 

6. We revised the abstract, the introduction and the conclusion. In Abstract Section, we added more details about how to validate our method, and our results. In Introduction Section, we modified the description about our method. In Conclusion Section, we added the future works of our method. 

7. We added more details about our experiment setup in Section 4.2 (Implementation Details).

8. Section 3 of the original manuscript was removed. 

Response to Reviewer 3: Thank you for your review of our paper. We reorganized the content of our paper and added more illustrations about our proposed method. Some inaccurate descriptions were also modified carefully. Besides, we also added new content including the process of image representation, the advantages and disadvantages and the future work. As for the experimental results, we are very 

The major changes are shown as follows: 

1. We added a new section (Section 3.1) named “Process of Image Representation”. In this section, we explained the five stages of the BoVW model in order, and illustrated which stage our method involves in. The key steps of our work were also given in order. A new figure (Figure 1) was presented in the revised manuscript, which shows the process of image representation including our method. 

2. We added a new section (Section 3.2) named “Observation Window”, which corresponds to the first step of our method. In this section, we introduced the principle of observation window. The advantages of using observation window were also stated. 

3. We added a new section (Section 3.3) named “Pooling Regions of Visual Word”, which corresponds to the second step of our method. At the beginning of this section, we illustrated the reason why the practice of making each word have its own pooling regions is feasible. Afterwards, we described how to learn its pooling region for each word. The learning process was divided into three small steps, which are illustrated in Section 3.3.1, Section 3.3.2 and Section 3.3.3, respectively. 

a) In Section 3.3.1 (Labelled Coordinate Data), we explained the reason why our method needs to build labelled coordinate data. Then, the details on building data were presented. 

b) In Section 3.3.2 (Pooling Region Learning), we first gave the entire learning process of pooling regions. Two key problems in the learning process are introduced in the subsection “Learning of Dividing Direction” and the subsection “Learning of the Best Dividing Line”. 

i. In the subsection “Learning of Dividing Direction”, we explained why LDA is used to learn a dividing direction. The details on how to learn dividing direction were also presented. 

ii. In the subsection “Learning of the Best Dividing Line”, the learning process of the best dividing Line was illustrated in detail. 

c) In Section 3.3.3 (Grouping the Responses of Observation Window), we illustrated how to group the responses of observation window by pooling regions. The advantage of using a part of all the pooing regions was also given. 

4. We added a new section (Section 3.4) named “Image Representation Vector”, which corresponds to the third step of our method. In this section, we explained how to obtain the image representation vector with the maximums from the groups of all the visual words. 

5. We added a new section (Section 4.7) named “Discussion”. In this section, the advantages and disadvantages of our method were illustrated. 

6. We revised the abstract, the introduction and the conclusion. In Abstract Section, we added more details about how to validate our method, and our results. In Introduction Section, we modified the description about our method. In Conclusion Section, we added the future works of our method. 

7. We added more details about our experiment setup in Section 4.2 (Implementation Details).

8. Section 3 of the original manuscript was removed.

Reviewer #3: 

1. please providing any inherent limitations and future directions of the proposed method.

Answer: A new section named “Discussion” was added in the revised manuscript. In this section, we illustrated the inherent limitations of our method, which are stated as follows. “The effectiveness of our method depends on inter and intra-class visual diversity to a great extent. When the spatial distributions of the local features relevant highly to image class are similar for the images from same class, our method works well. For example, the accuracy improvement of about 2% to 2.5% is obtained on Scene-15, Caltech-101 and Caltech-256. For most of the classes in the three datasets, the images are aligned artificially well. By contrast, only the gain of 0.9% is acquired on Corel-10 due to the large intra-class visual diversity.”

 We added the future directions in Conclusion Section, which are stated as follows. “The future works we are pursuing are: 1) learning the pooling regions of visual phrase; 2) applying the thought of our method on the convolution layers of CNN.”

---

## [Decision Letter · Decision Letter 1]

20 May 2020

Pooling Region Learning of Visual Word for Image Classification using Bag-of-Visual-Words Model

PONE-D-20-01860R1

Dear Dr. Xu,

We are pleased to inform you that your manuscript has been judged scientifically suitable for publication and will be formally accepted for publication once it complies with all outstanding technical requirements.

With kind regards,

Paweł Pławiak, D.Sc., Ph.D.

Academic Editor

PLOS ONE

Additional Editor Comments (optional):

Reviewers' comments:

Reviewer's Responses to Questions

**Comments to the Author**

1. If the authors have adequately addressed your comments raised in a previous round of review and you feel that this manuscript is now acceptable for publication, you may indicate that here to bypass the “Comments to the Author” section, enter your conflict of interest statement in the “Confidential to Editor” section, and submit your "Accept" recommendation.

Reviewer #1: All comments have been addressed

Reviewer #2: All comments have been addressed

Reviewer #3: All comments have been addressed

2. Is the manuscript technically sound, and do the data support the conclusions?

Reviewer #1: Yes

Reviewer #2: Yes

Reviewer #3: (No Response)

3. Has the statistical analysis been performed appropriately and rigorously? 

Reviewer #1: Yes

Reviewer #2: Yes

Reviewer #3: (No Response)

4. Have the authors made all data underlying the findings in their manuscript fully available?

Reviewer #1: Yes

Reviewer #2: Yes

Reviewer #3: (No Response)

5. Is the manuscript presented in an intelligible fashion and written in standard English?

Reviewer #1: Yes

Reviewer #2: Yes

Reviewer #3: (No Response)

6. Review Comments to the Author

Reviewer #1: The authors have addressed the reviewer's concerns and the revised version of the manuscript appears to be good.

Reviewer #2: The authors have improved the results and the steps of the work are well illustrated. I think all concerns have been satisfactorily addressed.

Reviewer #3: (No Response)

7. PLOS authors have the option to publish the peer review history of their article (what does this mean?). If published, this will include your full peer review and any attached files.

Reviewer #1: Yes: Mohamed Hammad

Reviewer #2: No

Reviewer #3: No

---

## [Editor Report · Acceptance letter]

26 May 2020

PONE-D-20-01860R1 

Pooling Region Learning of Visual Word for Image Classification using Bag-of-Visual-Words Model 

Dear Dr. Xu:

I am pleased to inform you that your manuscript has been deemed suitable for publication in PLOS ONE. Congratulations! Your manuscript is now with our production department. 

With kind regards,

on behalf of

Prof. Paweł Pławiak 

Academic Editor

PLOS ONE